# Historical Textile Dye Analysis Using DESI-MS

Edith Sandström [1,2,*], Chiara Vettorazzo [1], C. Logan Mackay [1], Lore G. Troalen [2] and Alison N. Hulme [1,*]

1   EaStCHEM School of Chemistry, University of Edinburgh, Edinburgh EH9 3FJ, UK
2   Department of Collections Services, National Museums Collection Centre, National Museums Scotland, Edinburgh EH5 1JA, UK
*   Correspondence: e.sandstroem@ed.ac.uk (E.S.); alison.hulme@ed.ac.uk (A.N.H.)

**Abstract:** Desorption electrospray ionization mass spectrometry (DESI-MS) is an ambient mass spectrometry technique that shows great potential for the analysis of fragile heritage objects in situ. This article focuses on the application of a recently built DESI source to characterize natural dyestuffs in historical textiles and a presentation of initial imaging experiments. Optimization of the instrumental settings, geometrical parameters, and solvent system on the DESI-MS analysis was conducted on rhodamine B samples. Some variables, including an increased flow rate, a narrower range of optimized geometrical variables and a solvent system without additives, were applicable to both early synthetic and natural dyes. Direct dye turmeric (*Curcuma longa* L.) could be reliably analyzed on both silk and wool, as could anthraquinone standards without mordanting. These preliminary results suggest that the dye application process (direct, mordant, vat) has a large impact on the ionization efficiency of DESI-MS. Imaging trials highlighted the suitability of DESI-MS for the analysis of patterned textiles that are difficult to sample, such as calico fabrics, or other currently inaccessible objects.

**Keywords:** mass spectrometry; non-invasive analysis; natural dyes; textile; mass spectrometry imaging

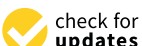



## 1. Introduction

The analysis of dyes in heritage objects, in particular fragile objects such as textiles, requires a continuous development of new analytical approaches that reduce the physical impact of analysis on the objects whilst ensuring that maximum information is gained. The common techniques used in dye analysis are either invasive, e.g., liquid chromatography (HPLC, UHPLC) and mass spectrometry (MS) [1], requiring a sample to be physically removed from the object, or non-invasive, e.g., fiber optics reflectance spectroscopy (FORS) [2] or Raman spectroscopy [3], meaning that the analysis has no physical impact on the object. The analytical techniques used may also be destructive or non-destructive, resulting in the sample being consumed during analysis or not. Often a combination of both approaches is preferred, with non-invasive methods guiding object sampling [4–6].

Ambient MS techniques, such as desorption electrospray ionization (DESI) [7], matrix-assisted desorption electrospray ionization (MALDESI) [8], and direct analysis in real time (DART) [9], are non-invasive albeit micro-destructive techniques with a large potential to be used in dye analysis. Desorption electrospray ionization (DESI)-MS was developed in 2004 by the Cooks research group at Purdue University [7]. It is an electrospray ionization (ESI)-based technique that is popular for its versatility and speed. Ionization occurs outside of the MS inlet by an electrospray of charged liquid droplets being directed onto the sample. Secondary charged droplets containing dissolved analyte are formed and desorbed from the sample surface and led into the MS inlet by voltages [10,11].

Textile fibers have previously been studied using DESI-MS in forensic and environmental contexts [12,13], and a DESI source was recently developed for the study of ink in manuscripts [14]. However, the first successful application of DESI-MS for the study

of historical dyes was only very recently reported by the authors of [15], focusing on the construction and optimization of a DESI source for early synthetic dye analysis. The work presented here is complementary to this study and highlights the initial investigations into the use of DESI-MS for the detection of natural dyestuffs and some preliminary imaging-type experiments facilitated by the XY stage set-up.

## 2. Materials and Methods

### 2.1. Materials

Alizarin, purpurin, carminic acid, and rhodamine B (CI 45170) were obtained from Sigma-Aldrich Inc., St. Louis, MO, USA. Fresh turmeric and turmeric powder were locally purchased. Samples of undyed, degummed, unmordanted silk (2-ply, 66 Tex, thread count 43 cm$^{-2}$) and undyed, washed, unmordanted wool cloth (3-ply, 158 Tex, thread count 36 cm$^{-2}$) were supplied by the Monitoring of Damage to Historic Tapestries project (MODHT) (FP5, EC contract number EVK4-CT-2001-00048) [16,17]. The H$_2$O, CH$_3$OH, and CH$_3$CN (LC-MS grade) were purchased from Fisher Scientific, Waltham, MA, USA. Fabric clips (Prym Love clips, $1.0 \times 2.6$ cm) and permanent markers (Lumocolor permanent pen 318, Staedtler, Nuernberg, Germany) were bought locally, while water-sensitive paper (Pentair Hypro) was purchased from Agratech NW Ltd., Rossendale, UK.

### 2.2. Dyeing Procedure

A mass of 100 mg $\pm$ 0.005 mg of each reference dyestuff was dissolved in 7.5 mL H$_2$O, and the dyebaths were heated to 75 °C before 100 mg $\pm$ 0.005 mg (ca. 1 cm$^2$) silk cloth was added. The dyebaths were kept at 75 °C for 15 min before the silk samples were removed and rinsed at least twice with cold deionized water and left to dry completely. The same dyebaths were used to dye the wool samples (150 mg $\pm$ 0.005 mg, ca. 1 cm$^2$) following the same procedure, except the wool samples were pre-wetted in deionized H$_2$O for 10 min before dyeing.

### 2.3. Instrumentation

A DESI source built in-house (Figure 1) attached to a Bruker 7T SolariX FT-ICR-MS using Compass HyStar 5.1 (Bruker Daltonik GmbH, Billerica, MA, USA) was used for all experiments. The commercial electrospray emitter (part number: 0601815, Bruker Daltonik) used for ESI-MS by the specified mass spectrometer was also utilized by the DESI source. The DESI source was constructed with an acrylic stage mounted on an XY stage controlled via an Arduino-compatible board (EleksMaker EleksMana V5.2) and LaserGRBL (v4.6.0). The sprayer holder was 3D printed in polylactic acid (Ultimaker 2, UltiMaker, Utrecht, Netherlands) and fitted onto a rotation mount (RP01 Manual Rotation Stage, THORLABS, Ely, UK) attached to three-direction positioners (Kite Manual Micromanipulator, WPI Inc., Sarasota, FL, USA). For spot monitoring, space for a camera (ESP32-CAM) was added above the sprayer tip. The sprayer set-up was mounted on stainless-steel rods attached to the stage and the whole assembly was affixed upon a lab jack for control of the sample-MS inlet height (*k*, Figure 1). The MS inlet was fitted with a $90 \times 0.4$ mm (length $\times$ i.d.) stainless-steel/brass extension, which was held in place around the MS capillary with a gold spring.

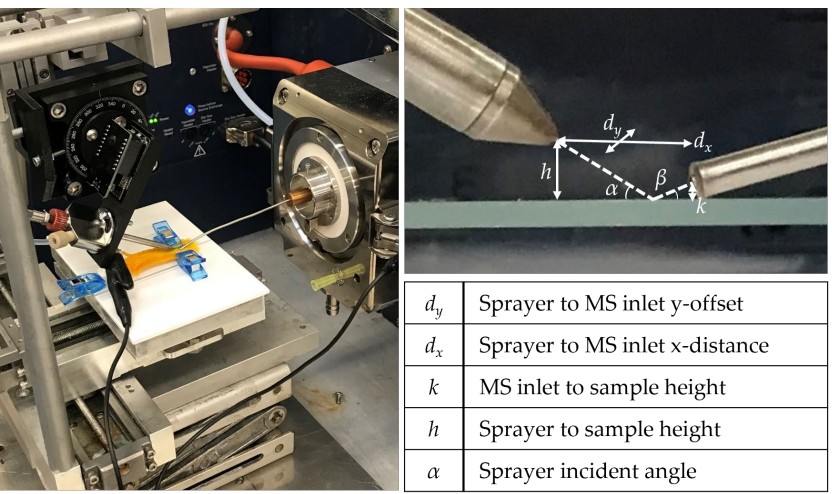

| $d_y$ | Sprayer to MS inlet y-offset |
|---|---|
| $d_x$ | Sprayer to MS inlet x-distance |
| $k$ | MS inlet to sample height |
| $h$ | Sprayer to sample height |
| $\alpha$ | Sprayer incident angle |

**Figure 1.** (**Left**): Photograph of constructed DESI source [15]. (**Right**): Detail of the sprayer and MS inlet of the constructed DESI source with key geometrical parameters. © University of Edinburgh/Edith Sandström.

### 2.4. DESI Analysis

Textile samples (ca. 1 cm$^2$) were placed on a glass slide and held in place with fabric clips (Prym Love clips, 1.0 × 2.6 cm). Larger textiles (ca. 10 × 10 cm) were clipped directly onto the plastic stage or, if large enough to not be affected by the nebulizing gas stream (ca. 25 × 25 cm), placed directly on the stage without clipping. DESI-MS spectra were acquired in the mass range of $m/z$ 150–1000 and 20 mass spectra were summed unless otherwise stated. The solvent system used was 3:1 $v/v$ CH$_3$CN:H$_2$O, and the MS inlet was cleaned with LC-MS grade CH$_3$CN and 3:1 $v/v$ CH$_3$CN:H$_2$O between every analysis. All spectra were processed using Compass DataAnalysis (Bruker Daltonik GmbH, Billerica, MA, USA) and Origin 9.5 (OriginLab, Northampton, MA, USA). The imaging investigation graph was constructed in GraphPad Prism 9.5.1 (GraphPad Software, LLC, San Diego, CA, USA).

The following parameters were optimized and used for the early synthetic dye samples in positive mode: capillary voltage: 4.5 kV, end plate offset: −500 V, flow rate: 750 μL h$^{-1}$, nebulizing gas: nitrogen, nebulizing gas pressure: 3.9 bar, source temperature: 200 °C, and negative mode: capillary voltage: 4.2 kV, end plate offset: −800 V, flow rate: 750 μL h$^{-1}$, nebulizing gas: nitrogen, nebulizing gas pressure: 3.9 bar, source temperature: 200 °C, sprayer angle: 35°, ion accumulation: 1.5 s. The parameters used for the early synthetic dyes in positive mode were also used in the imaging investigation.

Negative mode was used for all natural dye samples for the practical ease of changing between the natural dye classes tested. The following parameters were used for the turmeric root and powder: capillary voltage: 3 kV, end plate offset: −800 V, flow rate: 300 μL h$^{-1}$, nebulizing gas: nitrogen, nebulizing gas pressure: 1.5 bar, source temperature: 200 °C, sprayer angle: 76°, accumulation: 1.8 s. Parameters used for silk and wool samples dyed with turmeric: capillary voltage: 4.5 kV, end plate offset: −1000 V, flow rate: 1200 μL h$^{-1}$, nebulizing gas: nitrogen, nebulizing gas pressure: 4.0 bar, source temperature: 200 °C, sprayer angle: 76°, ion accumulation: 2.0 s.

The parameters used for the silk and wool samples dyed with alizarin and purpurin were as follows: capillary voltage: 3.9 kV, end plate offset: −800 V, flow rate: 500 μL h$^{-1}$, nebulizing gas: nitrogen, nebulizing gas pressure: 3.5 bar, source temperature: 250 °C, sprayer angle: 74°, accumulation: 1.8 s. The following parameters were used for the silk and wool samples dyed with carminic acid: capillary voltage: 3.9 kV, end plate offset: −800 V, flow rate: 750 μL h$^{-1}$, nebulizing gas: nitrogen, nebulizing gas pressure: 4.0 bar, source temperature: 250 °C, sprayer angle: 83°, ion accumulation: 1.8 s.

## 3. Results

### 3.1. Optimization of DESI Source for Historical Dye Analysis

DESI-MS is dependent on multiple interconnected parameters; instrumental, geometrical, and chemical in nature. It is especially dependent on the geometrical parameters of the sprayer set-up (Figure 1) [10,11,18–20]. The optimized geometry often differs with the target molecule and substrate used, making it difficult to standardize optimum geometry for all applications.

The initial testing of the geometrical, chemical, and instrumental ranges were all conducted using the absolute ion abundance of the rhodamine B peak $[M-Cl]^+$ (*m/z* 443.23), which found that the solvent system and sprayer angle had the largest effect on the ion abundance. The optimization of these two parameters and application of DESI-MS analysis to early synthetic dyes across six dye families on both silk and wool, has been published elsewhere [15]. Here, the effect and limitations of other parameters that we found to differ significantly for the analysis of textile samples compared to the analysis of more common surfaces, such as glass, are discussed in more detail, and the application of DESI-MS is extended to natural dyes and imaging.

The optimum solvent flow rate for DESI-MS has been shown to be highly dependent on the surface being analyzed [21–23]. In contrast to DESI-MS analysis of biological and forensic samples, higher flow rates were required when analyzing cloth. The greatest signal-to-noise ratio of the rhodamine B $[M-Cl]^+$ and background peaks was achieved at $750 \, \mu L \, h^{-1}$ [15], which is significantly higher than standard flow rates for biological and forensic samples ($90$–$300 \, \mu L \, h^{-1}$) [18,20,24]. The high flow rate required for textile samples has been reported previously [13], and it is likely needed due to the wicking properties of tightly woven cloths, reducing the formation of the thin liquid layer on the sample surface required for the proposed "droplet pick-up" mechanism for analyte transport [25]. Too low a flow rate resulted in the need for longer analytical times and gave a lower ion intensity, while higher flow rates ($1500$–$1800 \, \mu L \, h^{-1}$) resulted in the formation of water droplets on the cloth surface, particularly on wool (Figure 2).

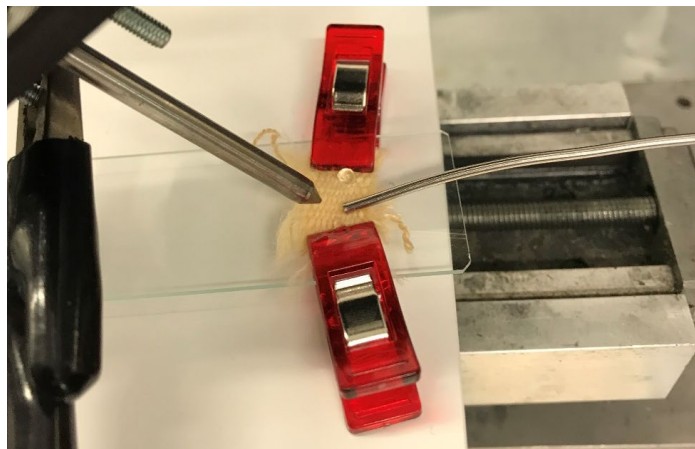

**Figure 2.** Undyed wool cloth being analyzed using too high a flow rate ($1800 \, \mu L \, h^{-1}$), resulting in droplet formation on the surface.

The sprayer–sample height (*h*, Figure 1) was found to be more limited when analyzing cloth compared to glass. Both cloth and glass showed higher ion abundances at lower heights. However, upon wetting, silk and wool become slightly conductive [26,27], which means that lowering the capillary too close to the sample resulted in burning of the cloth. This is an important difference to highlight, as any visual damage to the surfaces of objects needs to be avoided. The optimized height was *h* = 2 mm for most textile samples.

One of the major parameters investigated for the DESI-MS of textile samples was the sprayer angle, which has a direct influence on the spot area [11]. The effect of the

sprayer angle on the spot geometry was further investigated using solvent-sensitive paper (Figure 3). A flow rate of 750 µL h$^{-1}$, a sprayer-to-sample height (*h*, Figure 1) of 5 mm, and an analytical time of 1 min was used for each spot. The resulting spot areas were measured in ImageJ (Rasband, W. S., National Institutes of Health, Bethesda, MD, USA).

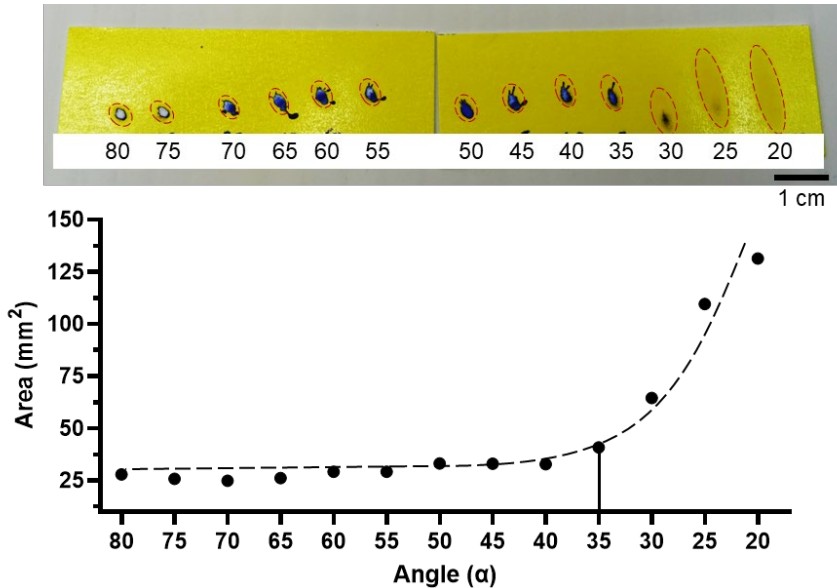

**Figure 3.** Solvent-sensitive paper (Pentair Hypro, Agratech NW Ltd., Rossendale, UK) showing the effect of different sprayer incident angles on the solvent (3:1 *v/v* CH$_3$CN:H$_2$O) spot shape and size. The solvent spot margins are marked by dotted lines. The measured area (mm$^2$) within the dotted lines is graphed below. The fitted curve is for data visualization only. An angle, *α*, of 35° is highlighted to show the critical point before the rapid increase in spot areas.

The spots which result from the use of higher angles show a more circular shape with a clear impact of the solvent in the center and fairly equal spot areas (Figure 3). As expected, the spot area increases, and the spot shape becomes more elongated, as the angle becomes shallower. Angles greater than 35° show similar spot areas across all solvent systems tested, while an exponential increase in spot area can be seen at angles 35°–20°.

The parameter values optimized on rhodamine B (listed geometrical parameters (Figure 1) and MS parameters (Appendix A, Table A1)) and tested on synthetic dye references, as well as historical early synthetic dye samples [15], were then applied to the analysis of natural dyes on textile samples and used with an imaging platform.

### 3.2. Natural Dyes

Turmeric (*Curcuma longa* L.) was chosen as a standard to determine the applicability of DESI-MS on natural dye chromophores based on it being a direct dye and readily available in both fresh and powder form. Initially, plant material was used to ensure that the DESI-MS set-up could ionize natural dye chromophores without interference from the cloth substrates. Fresh turmeric root purchased from a local grocery store was prepared in cryosections (50 µm), adhered to a glass slide, and analyzed using the DESI-MS set-up (Figure 4A). Additionally, turmeric powder was dissolved in H$_2$O, added to a glass slide, and left to evaporate overnight. The dried powder slides were then analyzed using DESI-MS (Figure 4B). Silk and wool cloth were also dyed with turmeric and analyzed by DESI-MS (Figure 4C,D).

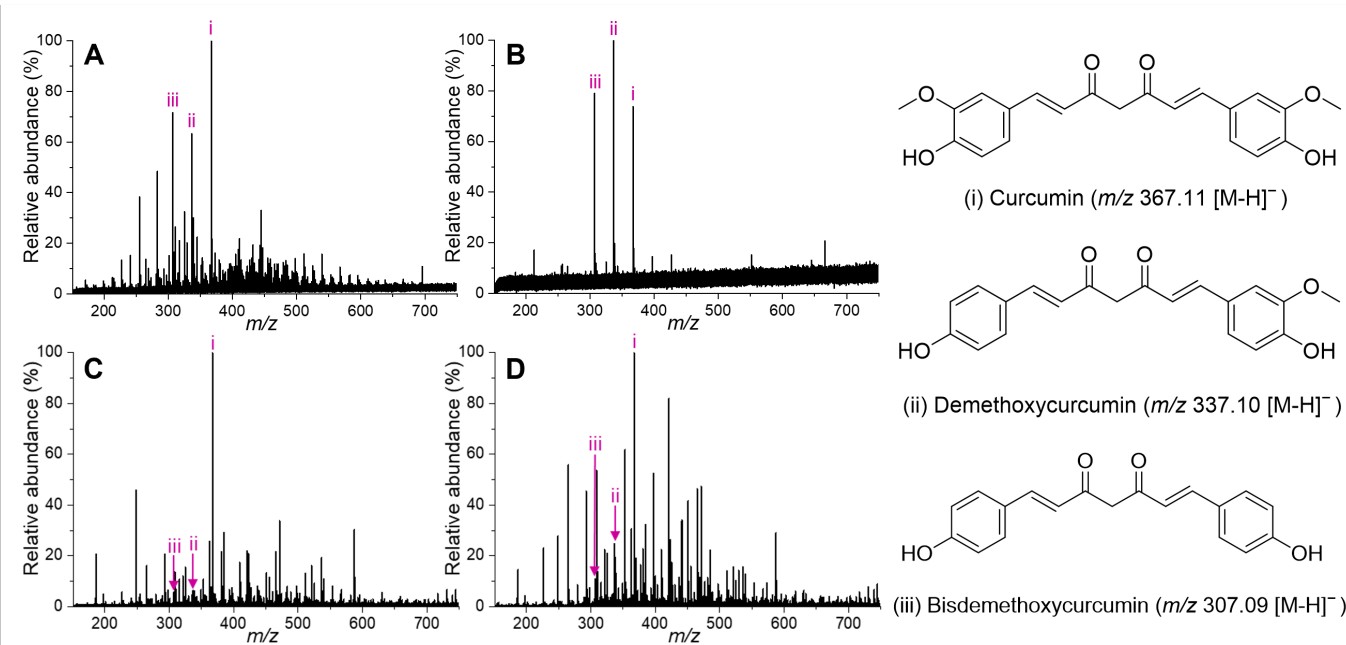

**Figure 4.** (**A**): Mass spectrum of 50 μm cryosections of fresh turmeric. (**B**): Mass spectrum of turmeric powder dissolved in $H_2O$ and left to evaporate on a glass slide. (**C**): Mass spectrum of turmeric dyed on silk cloth. (**D**): Mass spectrum of turmeric dyed on wool cloth. The three main dye components of turmeric (i)–(iii) are labeled in pink, and their $m/z$ and structure are shown on the right.

The three main dye components of turmeric—curcumin, demethoxycurcumin, and bisdemethoxycurcumin [28]—were all seen in the analysis of both the sectioned fresh and powder sample (Figure 4A,B). The mass spectrum obtained from the powder sample shows a less complex matrix than the fresh sample, likely due to processes during the drying step. Similar phenomena can be seen in the analysis of fresh and dried madder (*Rubia tinctorum* L.) [29], which is also a root used for dyeing. The noisier baseline in the turmeric powder spectrum (Figure 4B) compared to the fresh turmeric spectrum (Figure 4A) suggests that the chromophores are less concentrated in the powder sample. This change in concentration can be rationalized by the thinner sample layer on the glass slide, resulting in faster depletion of the analyte.

A signal from all three turmeric components could be seen on both the silk and wool cloth (Figure 4C,D), although these signals were less intense than they were for the fresh and powder turmeric samples (Figure 4A,B), and many background peaks could be seen. However, the signals were stable across the 20 mass spectra collected and showed a signal-to-noise ratio above 3:1, which is the common limit of detection used in the signal-to-noise method. The lower intensity peaks of turmeric compared to rhodamine B and the other early synthetic dyes tested show that the optimized set-up for early synthetic dyes [15] is not directly translatable to natural dyes and makes it likely that the lower ion abundance seen for the chromophores can be increased by optimizing the DESI parameters on a natural direct dye such as turmeric.

The silk and wool reference cloth dyed with mordant and vat dyes (weld (*Reseda luteola* L.), madder (*Rubia tinctorum* L.), cochineal (*Dactylopius coccus* Costa), and woad (*Isatis tinctoria* L.) for the MODHT project were investigated after the success of turmeric, but no expected $m/z$ peaks could be seen. One possible explanation for the lack of a signal for natural chromophores on the dyed cloth could be the mode of dye application, as all the samples showing no signal were mordant or vat dyes. A likely explanation for the limitation in the analysis of vat dyes is the known poor solubility of these dyes in aqueous systems [30]. Circumvention of this issue would require the addition of solvents such as DMSO or DMF; however, the potentially damaging effects of these additional solvents

on instrumentation, mass spectrum quality, and the objects themselves would need to be investigated.

This hypothesis was tested by dyeing unmordanted silk and wool cloth with standard references of anthraquinones (alizarin, purpurin, carminic acid) to see if any signal could be obtained from the silk and wool for common natural dye chromophores in the absence of mordant. The mass spectra obtained were promising as a high ion abundance of the [M-H]⁻ peak was observed for each of the compounds tested (Figure 5). All natural dye components were analyzed with a higher angle (75°) than optimized for rhodamine B to obtain any signal, highlighting the need for natural dye optimization. However, the acquisition of signals for common chromophores such as the anthraquinones on silk and wool is a promising start for the application of DESI-MS for textiles dyed with natural dyes, but the limitations seen for mordanted and vat dye molecules require further investigation.

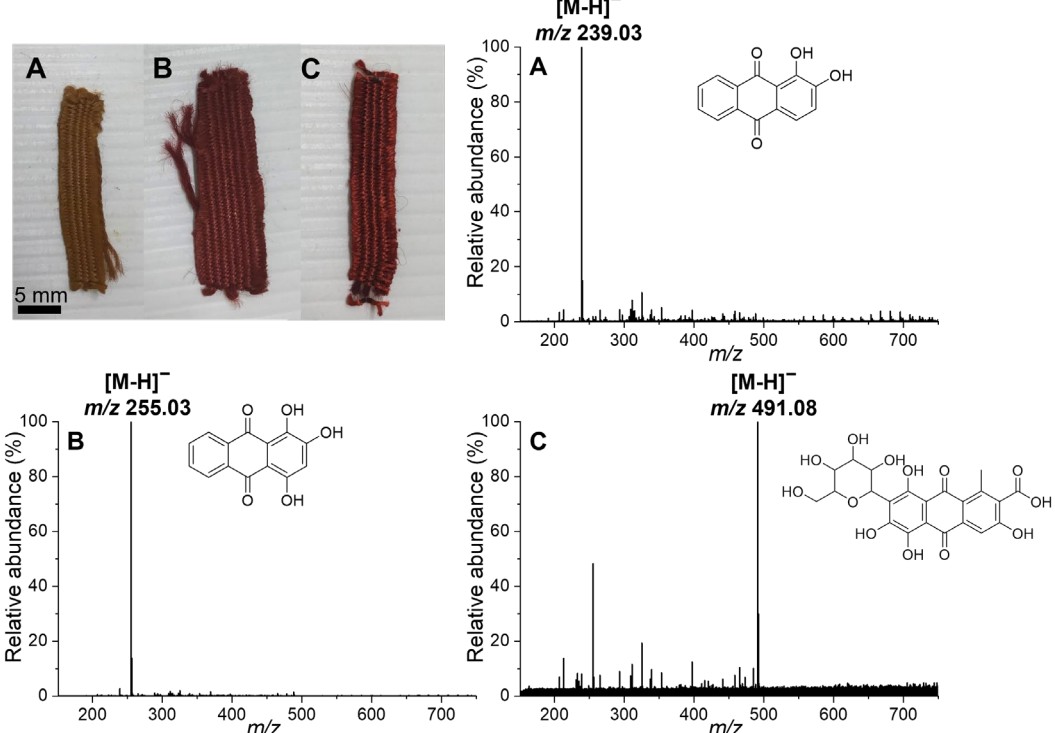

**Figure 5.** (**A**): Silk cloth dyed with alizarin, molecular ion labeled in mass spectrum A. (**B**): Silk cloth dyed with purpurin, molecular ion labeled in mass spectrum B. (**C**): Silk cloth dyed with carminic acid, molecular ion labeled in mass spectrum C.

### 3.3. Initial Imaging Experiments

One additional advantage of DESI is the possibility of mass spectrometry imaging (MSI) studies. Such an approach would be highly beneficial for the analysis of patterned textiles that are usually difficult to sample, such as printed calico and indienne fabrics. The DESI platform design included an electronically controlled stage for imaging purposes. Rhodamine B and a green ink (*m/z* 575.28) (permanent pen 318, Staedtler, Nuernberg, Germany) applied in stripes to a glass slide was used for the initial imaging investigations (Figure 6A).

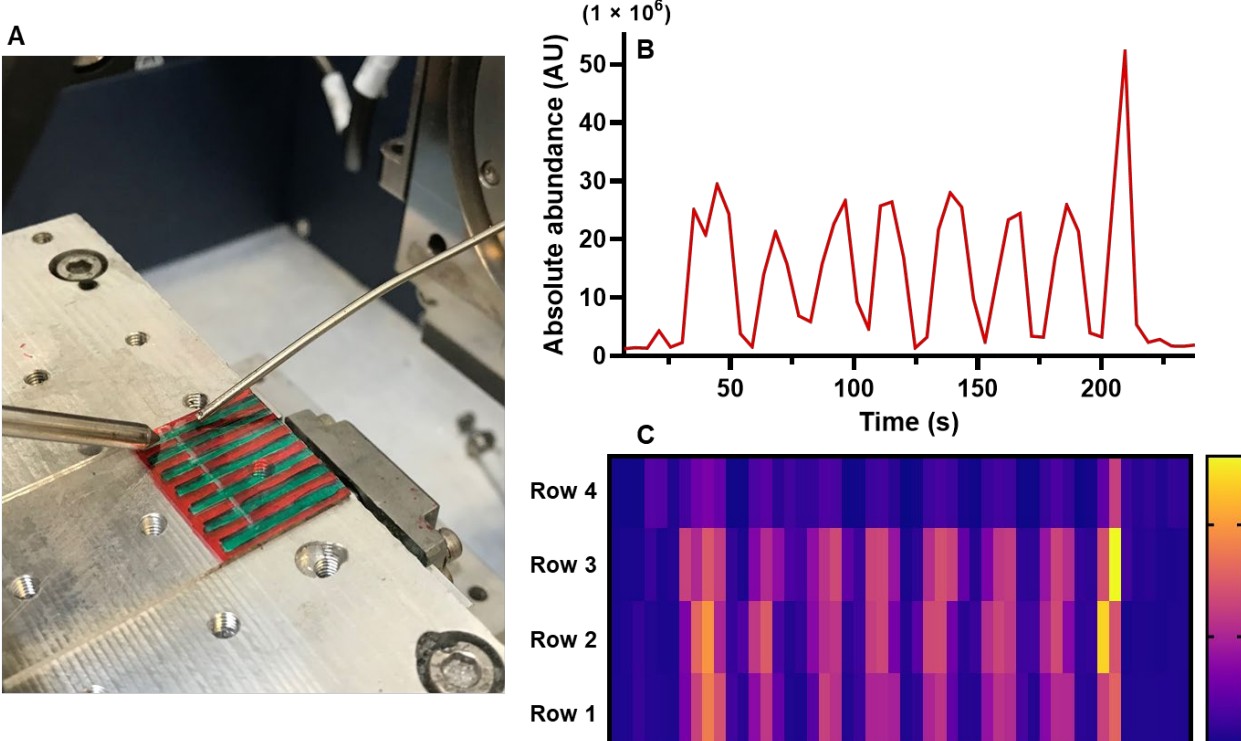

**Figure 6. (A)**: Photograph of imaging set-up with rhodamine B and a green ink in lines on a glass slide. **(B)**: Extracted ion chromatogram (*m*/*z* 443.23) recorded for row 3 showing the absolute ion abundance of rhodamine B recorded across the experimental time. **(C)**: Heat map constructed from the extracted ion chromatogram (*m*/*z* 443.23) of rhodamine B showing the lines of the painted design on the glass slide. Graph constructed in GraphPad Prism 9.5.1 (GraphPad Software, LLC, San Diego, CA, USA).

The stage was programmed to go across the stripes of rhodamine B and the green ink (*m*/*z* 575.28) (permanent pen 318, Staedtler, Nuernberg, Germany) at a speed of 8 mm min$^{-1}$, with each row being recorded individually. The extracted ion chromatogram for the rhodamine B peak (*m*/*z* 443.23) for each row (Figure 6B) was then exported, tabulated, and plotted as a heat map (Figure 6C). The successful imaging of rhodamine B on a glass slide and the reliable acquisition of *m*/*z* data for early synthetic dyes using the DESI set-up [15] suggest that MSI investigations of textiles patterned with early synthetic dyes are possible.

Some MSI investigations on textiles, in both archaeological and forensic settings, have been conducted [12,31,32] using matrix-assisted laser desorption/ionization time-of-flight (MALDI-TOF-MS), time-of-flight secondary ion mass spectrometry (TOF-SIMS), and infrared matrix-assisted laser desorption electrospray ionization (IR-MALDESI). The DESI-MS set-up developed shows a lower resolution but requires no sample preparation, in contrast to other MSI techniques. Thus, DESI-MSI could be of great use for the dye analysis field after the optimization of the spot to increase the spatial resolution and reduce the risk of cross-contamination.

The non-invasive imaging approach of DESI-MS offers an important method for the investigation of printed textiles, in which all of the colors present are often difficult to sample without threatening the structural integrity of the object. In the first instance, printed wool and silk fabrics, including pattern books and costumes, could be studied before moving to printed cotton textiles, such as calico and indienne fabrics [33]. Studies focused on cotton textiles would require further substrate-related optimization, including addressing considerations related to dye uptake and the strength of dye-fiber complexation. The imaging experiments conducted thus far will be followed up by further research into

spatial resolution optimization. However, these initial trials have shown that DESI-MS analysis is a promising method for the analysis of historical textiles and of great potential benefit for the dye analysis field.

## 4. Discussion

The analysis of historical dyes poses unique challenges, including low dye concentration, complex dye mixtures, and ethical considerations. The optimized DESI-MS set-up included a solvent system with no additives to reduce the analytical damage to the sample. It also required a higher flow rate and shallower sprayer angle than typically used on glass surfaces.

The optimized solvent system showed splattering of the solvent on both glass surfaces and the water-sensitive paper, even at low angles, resulting from the aqueous content and high flow rate used (Figure 3). This splattering is less significant when analyzing cloth due to wicking properties of the fabric. Splattering and the less precise spots of lower angles are disadvantages in imaging and any type of precision work, as they increase the risk of cross-contamination. However, lower angles and larger areas result in less damage to the spot investigated (Figure 3), as the same volume of solvent is spread over a larger area. This means that shallower angles provide an advantage when analyzing fragile heritage objects. This can be seen in the softer impact on the water-sensitive paper when the sprayer angle is at 20–30°. Such larger, less precise spots are desirable in the analysis of historical objects where imaging analysis is not the aim. The inclusion of a movable stage makes future application to imaging using the DESI source possible, after the optimization of the spot size and further investigations on the balance between spot damage and spatial resolution are conducted.

The difficulty in analyzing natural dyes, particularly mordant and vat dyes, highlights the fact that the parameters optimized for rhodamine B are not directly translatable, and there is a need for a similar optimization of natural dye standards. The successful analysis of individual chromophores directly applied to cloth suggests that DESI-MS is a promising technique for natural dye analysis, but further studies are needed to determine why there is a lack of ionization for mordant and vat dyes. Such studies need to focus on the difference in solubility in the aqueous solvent systems of natural dyes compared to early synthetic dyes and circumvent the stronger bonds within dye-mordant-fiber complexes. These issues would require the exploration of other solvent systems, which needs to be balanced against the impact of such systems on the object, making sure the analyses can still be considered non-invasive.

Future work will involve quantitative damage assessment to confidently apply the technique to culturally important textiles. The use of this in situ approach for dye analysis will be particularly significant to study textiles that cannot be sampled, such as block printed calico, indienne textiles, and also textiles that have undergone conservation treatments where the reverse is inaccessible for sampling.

Additionally, analyses of more complex systems, as well as unknown analytes, are underway to expand the limits of DESI-MS application to the field of dye analysis. The undoubted potential of in situ mass spectrometric analysis has been demonstrated, and future studies will hopefully expand its applications and give access to valuable information on hitherto inaccessible objects.

## 5. Conclusions

A DESI-MS source has been built in-house to allow for the non-invasive analysis of historical dyes. The set-up used the ESI sprayer attached to a Bruker 7T SolariX FT-ICR-MS instrument and included x-, y- and z-positioners, an angle mount for manual control of the geometrical parameters, and an electronically controlled stage. DESI-MS has been used successfully to analyze early synthetic dye references and historical samples using a higher flow rate and shallower sprayer angle compared to forensic and biological samples. The technique shows great potential for natural dye and imaging studies after further optimization.

**Author Contributions:** C.L.M., A.N.H., L.G.T., E.S. conceptualized the study. C.L.M., A.N.H., E.S., C.V. designed the experiments. E.S., C.V. carried out the experiments. E.S., C.V., C.L.M., A.N.H. analyzed the data. E.S. wrote the manuscript. A.N.H., L.G.T., C.L.M., C.V. read and edited the manuscript. All authors have read and agreed to the published version of the manuscript.

**Funding:** This research was funded by Scottish Cultural Heritage Consortium AHRC CDP (AH/S00176X/1 Studentship to ES).

**Data Availability Statement:** The data presented in this study are openly available at https://datashare.ed.ac.uk/handle/10283/4853.

**Acknowledgments:** We thank Lauren Ford (Imperial College London, UK) for fostering helpful discussions. We also thank the Monitoring of Damage to Historic Tapestries (MODHT) project for silk and wool cloth samples (EC contact: EVK4-CT-2001-00048).

**Conflicts of Interest:** The authors declare no conflict of interest.

## Appendix A

**Table A1.** MS parameters tested for their effect on the absolute ion abundance of Rhodamine B $[M-Cl]^+$ peak ($m/z$ 443.23), the range tested, and the optimized ranges.

| MS Parameter | Range Tested | Range for Greatest Absolute Ion Abundance ($m/z$ 433.23) (n = 3) |
|---|---|---|
| Sweep excitation energy | 12–15% | 15% |
| Skimmer voltage | 5–30 V | 15–25 V |
| Source temperature | 200–300 °C | 250–300 °C |
| Sprayer capillary voltage | 1–5 kV | 4–4.5 kV |
| Dry gas flow | 1.0–5.0 bar | 3.5–4.0 bar |
| Accumulation | 0.2–3.0 s | 1.5–2.5 s |
| Time-of-flight (ToF) | 0.2–0.8 ms | 0.5–0.7 ms |
| Flow rate | 200–2000 μL h$^{-1}$ | 700–800 μL h$^{-1}$ |

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
