# Peer review of "Historical Textile Dye Analysis Using DESI-MS"

_heritage, doi:10.3390/heritage6050212_

Round 1

Reviewer 1 Report

The article presents a non-invasive ambient technique that opens a new horizon for the analysis of dyes and pigments in the field of cultural heritage. The manuscript itself reads very well and it is well structured. However, I am concerned about the novelty of the article as the same authors have recently published a very similar paper in Analytical Chemistry (https://doi.org/10.1021/acs.analchem.2c03281).

Result section 3.1 has been already published in the AC articled and there is no need to include it again in this paper, e.g., figures 1 and 5 are modified versions of the AC article and the optimization of all parameters has been already described in the AC paper. Section 3.1 can be reduced to the last sentence of the section: "The parameter values presented were all optimized on rhodamine B and tested on synthetic dye references as well as historical early synthetic dye samples[27]. Here we report the extension of this study to natural dyes and imaging"

I would encourage to the authors to elaborate/extend more sections 3.2 and 3.3 initial imaging experiment (which actually is labeled as 3.1-please correct), as the new results are worth of publication and they can make an impact in the field.

Reviewer 2 Report

The paper presents a very interesting study regarding the optimization of a DESI-MS procedure for the analysis of synthetic and natural dyes highlighting the potentialities and the limits of the proposed methods.

The study is very well structured, thoroughly discussing all the parameters evaluated for the fine-tuning of the protocol. The work results are adequately referred, very well written, contextualized and properly self-critical.

The scientific results collected, and the considerations drawn highlight the potentialities of this approach to be applied for future studies.

Only few comments:

Line 38: remove “by these techniques”.

Line 173-174: include full names related to chemical formula.

Line 318: remove “would be”.

Reviewer 3 Report

This is a very nice paper that could benefit from some very minor changes as suggested in the attached pdf.  DESI has known issues with solubility, and that could be emphasized more, along with the problems that mordants induce in removing the dye from the fibers.  Overall, great study!

Round 2

Reviewer 1 Report

I really appreciate the effort that the authors have done to change the manuscript according to the comments.